# Bioaccumulation of Trace Elements along the Body Longitudinal Axis in Honey Bees

Enzo Goretti [1] , Matteo Pallottini [1,*] , Gianandrea La Porta [1] , Antonia Concetta Elia [1] , Tiziano Gardi [2],
Chiara Petroselli [1] , Paola Gravina [1] , Federica Bruschi [1] , Roberta Selvaggi [1] and David Cappelletti [1]

1   Dipartimento di Chimica, Biologia e Biotecnologie, Università degli Studi di Perugia, Via Elce Di Sotto 8, 06123 Perugia, Italy; enzo.goretti@unipg.it (E.G.); gianandrea.laporta@unipg.it (G.L.P.); antonia.elia@unipg.it (A.C.E.); chiara.petroselli@unipg.it (C.P.); paolagravi@gmail.com (P.G.); federica.bruschi@studenti.unipg.it (F.B.); roberta.selvaggi@unipg.it (R.S.); david.cappelletti@unipg.it (D.C.)

2   Dipartimento di Scienze Agrarie, Alimentari ed Ambientali, Università degli Studi di Perugia, Borgo XX Giugno 74, 06121 Perugia, Italy; tiziano.gardi@gmail.com

*   Correspondence: pallottini.matteo@gmail.com

**Abstract:** We present a survey on the environmental contamination of the Alviano Lake territory (Central Italy) based on *Apis mellifera ligustica* samples collected in two annual samplings (2019–2020). Concentrations of 30 elements were determined in the whole bees, in the gaster, and in the body without the gaster. The study generally revealed a low level of contamination of the bee tissues. However, As showed higher concentrations than in other rural areas, although lower than in samples from urban and productive areas. On the other hand, despite the environmental context, Hg showed limited contamination levels, with the exception of a single sample. Elemental analysis along the longitudinal axis of the bees' bodies showed greater and statistically significant presences of V, Al, Be, Pb, Cd, Co, Mn, Ba, and Sr in the gaster. The only exceptions concerned As and S (and to a lesser extent Hg), with higher concentrations found in the body without the gaster. We hypothesise that this selectivity maybe due to the affinity of these elements with S, which is abundant in the proteins of the flight muscles in the insect thorax, which are rich in amino acids containing the –SH group.

**Keywords:** pollution; metals; bioindicator; *Apis mellifera ligustica*; Honeybee Contamination Index; gaster

## 1. Introduction

Honey bees, *Apis mellifera* Linnaeus, 1758, primarily *Apis mellifera ligustica* (Spinola, 1806), are frequently used as bioindicators of the terrestrial environment quality—being very sensitive to environmental pollution caused by urbanisation, industrialisation, and agriculture—of all environmental compartments (i.e., air, soil, water, and vegetation) [1–7]. These pollinator insects are widespread in the territory due to the practice of beekeeping. Their colonies reside in a specific zone, usually frequenting a limited area of about 7 km$^2$ with a radius of about 1.5 km from the hive [7,8]. The foraging bees of a single hive (about 10,000 specimens, a quarter of the total colony population) during the spring–summer period visit up to 10 million flowers per day [9,10] and consequently intercept a large number of pollutants in all the environmental compartments during their large-scale foraging behaviour. Therefore, the level of contaminant bioaccumulation in their tissues and the hive products is an effective detector of environmental contamination [10–13].

Honey bees are susceptible to pesticides. Pesticide exposure effects can be acute and lethal (rapid and severe, quickly leading to death) or sub-acute and sub-lethal, not causing mortality. In the first case, the number of dead bees in front of the hive is the primary variable to be considered [2]. In the second case, the physiological or long-term behavioural effects [14,15] must be considered. There has been much talk about colony collapse disorder (CCD) in recent years. This phenomenon, which consists of the death

of bees with the loss of entire colonies or depopulation of many apiaries, appeared in 2005/2006. According to some authors, one of the main responsible factors is the use of neonicotinoid pesticides [16,17]. In addition, bees weakened by insecticide poisoning are more susceptible to pests and diseases [18–21].

Contamination by metals can be distinguished from that by pesticides due to their different environmental fates. Metals do not decay in the environment and are characterised by peculiar latent toxicity [2]. Metals generally are accumulated in the honeybee tissues, thus making honeybees efficient bioindicators [11,22,23]. Therefore, honey bees can be exploited to monitor metal pollution, usually caused by anthropic activities, which allows highlighting of the areas of most significant environmental risk with repercussions not only for the health of ecosystems but also for human health [24]. In addition, due to the relatively short lifetime of these insects, it is possible to verify the validity of any environmental recovery strategies in use in the territory [12]. However, the relationship between the levels of the elements in the environment and biota is very complex. It is based on their bioavailability, which presupposes the elements' entry into the food chain and their assimilation by organisms into their tissues [25]. Organisms require essential elements, but excess amounts can cause the onset of diseases. Some elements are instead toxic at any concentration (i.e., As, Cd, Hg, and Pb) and not required for any metabolic function [26–28].

This study's main objective is to analyse the accumulation levels of trace elements along the body axis of honey bees. We hypothesise that the tagmata of the bee body should have a diversified level of accumulation of trace elements. In particular, the abdomen should show the highest concentrations because it contains the excretory system (Malpighian tubule system), which is connected to the intestine right in the area between the midgut and the hindgut, and the rectum, which is the place of storage and disposal of waste. In order to verify this hypothesis, we considered, as a case study, the bioaccumulation of trace elements in the tissues of honey bees collected at the artificial lake of Alviano (Central Italy). Alviano Lake is a Natura 2000 Site (SAC IT5220011 and SPA IT5220024) and a naturalistic oasis of the World Wide Fund for Nature (WWF). This wetland is also part of the Tiber River Regional Park (Umbria Region). It is an area of particular naturalistic value, subject to low environmental impact. It is about 740 hectares in size, with shallow waters, marshes, hygrophilous woods, and cultivated areas. However, this territory suffers from mercury contamination [29] because the catchment area of Alviano Lake includes the Monte Amiata mining district (about 60 km away). This area is drained by the Paglia River, a tributary of the Tiber River, which enters just upstream of the barrier of the Alviano reservoir. This mining district was active for about 100 years, until 1980, and represented the third largest district in the world for Hg production. For this reason, the Paglia–Tiber River system is considered one of the main contributors to the total mercury balance in the Mediterranean Sea, which represents 65% of the world's mercury [29,30].

Therefore, this study has the following aims: (i) to identify the levels of trace element accumulation along the body axis of honey bees, using them as an excellent model to assess ecological and health risks, and (ii) to investigate mercury contamination in these pollinating insects.

## 2. Materials and Methods

### 2.1. Study Area

Lake Alviano is an artificial basin (altitude 77.5 m above sea level and area of 3.49 km$^2$) built in 1963 on the Tiber River, just downstream of the confluence of the tributary Paglia River, with a maximum initial depth of about 11 m at full capacity. The swamping created an artificial wetland consisting of a shallow lake of about 500 hectares, with a medium depth of about 30 cm [31–33]. The dam was built for electricity production and as an expansion tank for the Tiber River floods to safeguard the downstream cities, in particular Rome. In a short time, the sediments have given rise, since 1977, to this important and protected wetland.

The Paglia River drains the eastern side of Mount Amiata, formerly the third leading world district for mercury production, with about 100,000 tons produced in over 100 years of mining activity (mid-1800s–1980) [29]. According to different studies [30,34–39], at least 30,000 tons of Hg were dispersed into the environment, notably into the hydrographic basin of the Paglia River, thus contaminating the Alviano Lake area.

### 2.2. Honey Bee Sampling Campaign

The survey was carried out in 2019 and 2020 at Alviano Lake (sites 1 and 2), with two annual honey bee samplings during the warm season (July and September 2019, as well as June and September 2020) (Figure 1). Three hives (a, b, and c) were sampled at each of the two Alviano sites (site 1: 70 hives; site 2: 64 hives), taking care to select hives distant from each other (one in the centre and the other two at the ends of the apiary).

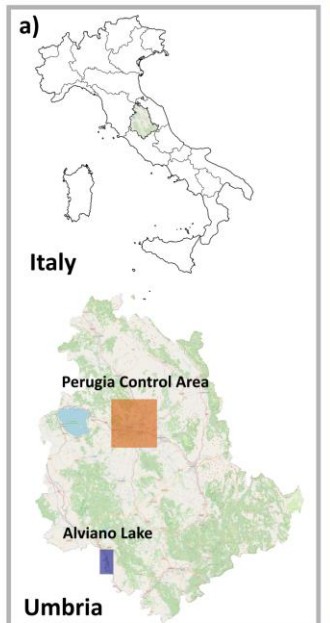 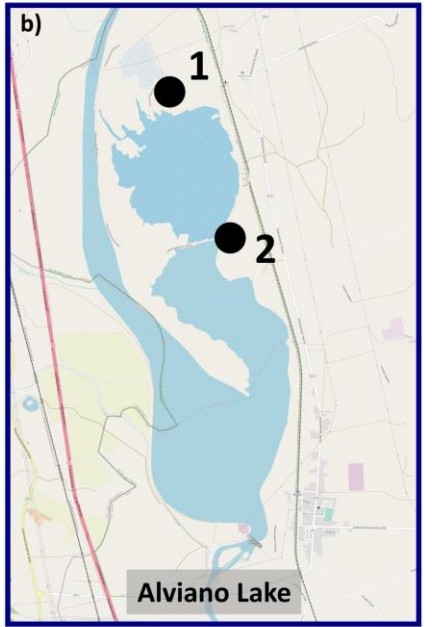 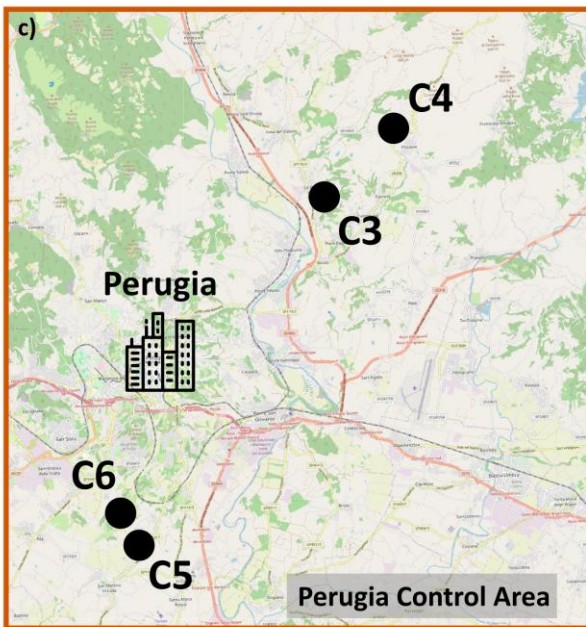

**Figure 1.** Map of the study area. (**a**) Localization of the two sampling areas in the Umbria Region, Central Italy, with the blue rectangle indicating Alviano Lake and the orange rectangle indicating the Perugia control area; (**b**) Alviano Lake, sites 1, 2; and (**c**) Perugia control area, sites C3, C4, C5, C6.

As a control, an area near Perugia was chosen (Central Italy) from among those that in a previous study (2014–2015, [40]) showed the lowest metal contamination levels in the Umbria region. Four control sites were selected (site C3: Ramazzano, site C4: Montelabate, site C5: San Fortunato della Collina, and site C6: Boneggio). These sites were sampled in September 2020, and only one central hive was sampled at each control site (Figure 1).

Therefore, for each sample is indicated the site number: 1, 2 (Alviano Lake sites), C3, C4, C5, and C6 (control area (Perugia) sites); the different hives analysed at each site: a, b, and c; and the month and year of sample collection.

About 100 forager bees were collected at the entrance of each hive with a plastic bag. Honey bee sampling was carried out according to good beekeeping practices, avoiding interference with the colony activity through opening the hives. All the apiaries sampled were characterized by treatment protocols compatible with organic beekeeping practices.

### 2.3. Metal Analysis

Honey bees were refrigerated on site. In the laboratory, the samples were stored at a temperature of −20 °C. Before further processing, drones, if present, were removed from the samples, and then each honey bee specimen was accurately cleaned with ultrapure water to remove any pollen that could be present in the pollen baskets of the hind legs, any other

particles present on the body surface, and any parasites, i.e., *Varroa destructor* (Anderson and Trueman, 2000). After the specimens had dried, for each sample, 15 bees were processed as whole bodies (sample name: "B"), and 30 bees were processed after separation of the anterior portion of the body, including the head, thorax, and first abdominal segment (sample name: "B−G"), from the remaining abdominal segments (sample name: "G") via cutting with ceramic scissors at the level of the petiole.

All the samples were subjected to freeze-drying under vacuum conditions at a temperature of −60 °C for 72 h. Once the samples were freeze-dried, they were placed in a desiccator until acid digestion was carried out. The samples were weighed with a precision balance, up to a maximum of about 0.5 g. They were processed with 8 mL of sub-boiling distilled 67–69% ultrapure NORMATON $HNO_3$ and heated up to 180 °C (15 min of heating up to a temperature of 170 °C, 30 min of holding at 170 °C, and 15 min of cooling) in a microwave digester (MARS 6, CEM, Matthews, NC, USA). After cooling, the digested samples were transferred quantitatively into Falcon tubes, and 2 drops of ultrapure HCl were added to each sample for mercury stabilisation. Then, ultrapure water was added to the samples to reach a volume of 25 mL.

Concentrations of 30 elements (Ag, Al, As, B, Ba, Be, Ca, Cd, Co, Cr, Cs, Cu, Fe, Ga, Hg, K, Li, Mg, Mn, Na, Ni, P, Pb, Rb, S, Si, Sr, Tl, V, and Zn) in the samples were determined with an inductively coupled plasma mass spectrometer (ICP-QQQMS, 8900 Agilent, Santa Clara, CA, USA). Commercially produced standard solutions (ICP multi-element standard solution CertiPUR®, 1000 mg $L^{-1}$, Merck Chemicals and Reagents, Darmstadt, Germany) in nitric acid and internal standard solution (internal standard mix for ICP-MS systems, 100 mg $L^{-1}$, 6-Li, Sc, Ge, Rh, In, Tb, Lu, Bi, Agilent Technologies Italia S.p.a., Milan, Italy) were used to prepare appropriate elemental calibration standards. An ordinary least-squares regression model was used for calibrations. Linearity between counts per second (CPS) and concentration (R2 > 0.999) was observed for Hg in the range of 0.01–10 ppb and in the range of 0.1–100 ppb for the other metals. The digested solutions were diluted at a 1:10 ratio using ultrapure water (18 MΩ). Experimental repeatability was calculated by performing three replicate analyses of two multi-element standard solutions (0.5 and 1 μg $L^{-1}$). The metal RSDs obtained by the repeatability test were good and were in the range of 2.6–13.0%. The accuracy of the method was obtained using standard reference materials (BCR-185R bovine liver). The metal concentrations agreed with the certified values, and recovery fell in the range of 80–120% [40]. Analytical quality control included analysis of a digestion reagent blank with each batch of 16 samples. All laboratory glassware was soaked in 10% $HNO_3$ for 24 h and then rinsed before use with ultrapure water [41–43]. The ICP measures were replicated three times for each sample. The detection limits of the method (in mg $kg^{-1}$) were: Ag, 0.0002; Al, 0.08; As, 0.001; B, 0.10; Ba, 0.001; Be, 0.0001; Ca, 0.28; Cd, 0.0002; Co, 0.0005; Cr, 0.003; Cs, 0.0008; Cu, 0.001; Fe, 0.01; Ga, 0.0006; Hg, 0.0007; K, 0.38; Li, 0.09; Mg, 0.03; Mn, 0.003; Na, 0.18; Ni, 0.0004; P, 0.04; Pb, 0.002; Rb, 0.001; S, 0.04; Si, 0.05; Sr, 0.001; Tl, 0.0001; V, 0.0005; and Zn, 0.09. All concentrations were expressed on a dry weight basis (dw).

*2.4. Honeybee Contamination Index*

The Honeybee Contamination Index (HCI) allows the evaluation of the metal contamination level in honey bee tissues (*Apis mellifera ligustica*) using the combination of the HCI values calculated for two contamination thresholds: threshold$_1$ (higher) and threshold$_2$ (lower). A positive HCI$_1$ indicates a high contamination level, a negative HCI$_2$ indicates a low contamination level, and finally, when HCI$_1$ is negative and HCI$_2$ is positive, an intermediate contamination level is present [40].

Equation:

$$HCI_1 = Log10 \ ([bees]/threshold_1); \ HCI_2 = Log10 \ ([bees]/threshold_2);$$

$$HCI_{mean1} = mean \ HCI_1; \ HCI_{mean2} = mean \ HCI_2$$

where [bees] is the element concentration in honey bee tissues; threshold$_1$ and threshold$_2$ are the element maximum and minimum concentration thresholds in honey bee tissues, respectively; and HCI$_{mean}$ is the mean of the index calculated for all the considered elements.

The HCI is only applicable to the samples of whole honey bee bodies (B) and for the elements for which threshold values are known in the literature [5,12,44], i.e., Cd, Pb, Cr, and Ni; their thresholds are 0.1, 0.7, 0.12, and 0.3 mg kg$^{-1}$ for threshold$_1$ and 0.05, 0.3, 0.04, and 0.1 mg kg$^{-1}$ for threshold$_2$, respectively. The conversion of these values from wet to dry weight values was made considering a weight loss of 68% due to the drying process [45].

### 2.5. Data Analysis

The database has been managed with descriptive analysis of the individual dataset variables. The Mann–Whitney U test was used to test for significative differences ($p > 0.05$) among element concentrations in honey bees among the hives sampled at the two Alviano sites, between the two Alviano sites, between Alviano and the control sites, and between gaster samples and body samples without gaster. Statistical analyses were conducted using the R statistical framework [46].

## 3. Results

The survey at Alviano Lake (sites 1, 2), carried out in 2019 and 2020, was based on 72 honey bee samples (24 for each sample type, i.e., B, B−G, and G) collected on four sampling occasions at three different hives for each site.

The element concentrations in the whole bee tissues (body) of Alviano Lake were not significantly different (Mann–Whitney U test, $p > 0.05$) among the three hives (a, b, and c), both at site 1 and site 2. No significant differences were detected also between the two sites (1, 2). As a result, all the Alviano samples were subsequently processed comprehensively as Alviano Lake area samples.

At the four control sites (sites C3–C6), located close to the city of Perugia (Umbria, Central Italy), 12 samples (4 for each sample type, i.e., B, B−G, and G), collected in September 2020, were processed comprehensively as Perugia Control area samples.

The mean concentrations of the 30 trace elements (single values, means, and standard deviations are reported in the Supplementary Material, Table S1a–c) for the Alviano and control areas were grouped into 8 classes (with a variation of a 10 factor), from concentrations lower than 0.001 mg kg$^{-1}$ to concentrations higher than 1000 mg kg$^{-1}$. This analysis was carried out separately for the three different sample types, i.e., whole honey bee body (B), abdominal segments after the petiole (G), and the remaining part of the body (B−G) (Table 1a,b).

The trace elements showing the lowest mean concentrations (class < 0.001) in Alviano honey bee tissues were, for B and B−G, Be (0.0008 mg kg$^{-1}$ and 0.0002 mg kg$^{-1}$, respectively), and Ag (0.0008 mg kg$^{-1}$ and 0.0009 mg kg$^{-1}$, respectively), while for gaster the elements Ag (0.002 mg kg$^{-1}$) and Be (0.002 mg kg$^{-1}$) were in the superior class (0.001–0.01), together with Hg (0.007 mg kg$^{-1}$) and Ga (0.009 mg kg$^{-1}$). In the control area, the first class (<0.001) was always empty for gaster tissues, while the lowest values were of Tl (0.0004 mg kg$^{-1}$) and Be (0.0009 mg kg$^{-1}$) for B, and Tl (0.0002 mg kg$^{-1}$), Be (0.0004 mg kg$^{-1}$), Ag (0.0005 mg kg$^{-1}$), and Cs (0.0008 mg kg$^{-1}$) for B−G.

On the other hand, the trace elements in Alviano showing the highest levels in honey bee tissues (class >1000) for B and B−G were S (2625.77 mg kg$^{-1}$ and 3087.50 mg kg$^{-1}$, respectively), P (4157.22 mg kg$^{-1}$ and 4055.21 mg kg$^{-1}$, respectively) and K (4407.09 mg kg$^{-1}$ and 4259.24 mg kg$^{-1}$, respectively), while for gaster tissues these elements (S, 2153.56 mg kg$^{-1}$; P, 5055.63 mg kg$^{-1}$; and K, 5598.33 mg kg$^{-1}$) were accompanied by Ca (1200.03 mg kg$^{-1}$). The conditions in the control area sites were similar. The elements showing the highest levels (class >1000) for B, G, and B−G were S (2323.33 mg kg$^{-1}$, 1875.13 mg kg$^{-1}$, and 2649.64 mg kg$^{-1}$, respectively), P (3859.63 mg kg$^{-1}$, 4565.84 mg kg$^{-1}$,

and 3675.06 mg kg$^{-1}$, respectively), and K (3964.07 mg kg$^{-1}$, 4294.02 mg kg$^{-1}$, and 3829.48 mg kg$^{-1}$, respectively), for gaster tissues; also, Ca is included (1317.58 mg kg$^{-1}$).

**Table 1.** Analysis of the mean concentrations (mg kg$^{-1}$, dry weight) of the 30 trace elements for (**a**) Alviano Lake sites and (**b**) Perugia control area sites. The mean concentrations are organised into 8 classes (varying by factors of 10), from concentrations lower than 0.001 mg kg$^{-1}$ to concentrations higher than 1000 mg kg$^{-1}$. This analysis was carried out separately for the three different sample types, i.e., whole honey bee body (B), abdominal segments after the petiole (G), and the remaining part of the body (B−G).

**(a) Alviano Lake**

| mg kg$^{-1}$ | BODY (B) | | | | GASTER (G) | | | | BODY without GASTER (B−G) | | | |
|---|---|---|---|---|---|---|---|---|---|---|---|---|
| <0.001 | Be 0.0008 | Ag 0.0008 | | | | | | | Be 0.0002 | Ag 0.0009 | | |
| 0.001–0.01 | Ga 0.003 | Tl 0.005 | Hg 0.006 | Cs 0.007 | Ag 0.002 | Be 0.002 | Hg 0.007 | Ga 0.009 | Ga 0.001<br>V 0.006 | Cs 0.002<br>Hg 0.006 | Cd 0.003<br>Co 0.009 | Tl 0.004 |
| 0.01–0.1 | V 0.023<br>Co 0.092 | Cr 0.026<br>Li 0.094 | Cd 0.028 | Pb 0.050 | Tl 0.011<br>As 0.077 | Cs 0.020<br>Cd 0.084 | Cr 0.056 | V 0.064 | Cr 0.010<br>Sr 0.070 | Pb 0.014<br>Li 0.095 | Ni 0.035 | Ba 0.058 |
| 0.1–1 | As 0.105 | Ni 0.120 | | | Li 0.107 | Pb 0.182 | Ni 0.265 | Co 0.269 | As 0.130 | | | |
| 1–10 | Ba 1.51 | Sr 2.33 | Rb 4.20 | B 6.97 | Ba 5.08 | Rb 6.87 | Sr 7.52 | | Mn 2.05<br>Cu 7.50 | B 2.61<br>Si 9.02 | Al 2.79 | Rb 3.32 |
| 10–100 | Al 10.71<br>Zn 51.85 | Cu 11.05<br>Fe 77.82 | Si 26.05 | Mn 48.82 | B 17.05 | Cu 19.19 | Al 29.21 | Si 65.76 | Zn 25.32 | Fe 52.82 | | |
| 100–1000 | Na 319.60 | Mg 455.41 | Ca 494.26 | | Zn 117.12<br>Mg 589.69 | Fe 149.08 | Mn 154.78 | Na 432.11 | Ca 197.08 | Na 295.37 | Mg 422.60 | |
| >1000 | S 2625.77 | P 4157.22 | K 4407.09 | | Ca 1200.03 | S 2153.56 | P 5055.63 | K 5598.33 | S 3087.50 | P 4055.21 | K 4259.24 | |
| **(b) Control Area (Perugia)** | | | | | | | | | | | | |
| mg kg$^{-1}$ | BODY (B) | | | | GASTER (G) | | | | BODY without GASTER (B−G) | | | |
| <0.001 | Tl 0.0004 | Be 0.0009 | | | | | | | Tl 0.0002 | Be 0.0004 | Ag 0.0005 | Cs 0.0008 |
| 0.001–0.01 | Ag 0.001 | Hg 0.002 | Cs 0.003 | Ga 0.005 | Tl 0.001<br>Cs 0.007 | Be 0.002 | Hg 0.004 | Ag 0.005 | Ga 0.001<br>Co 0.008 | Hg 0.002 | Cd 0.004 | V 0.007 |
| 0.01–0.1 | As 0.028<br>Pb 0.061 | Cr 0.029<br>Li 0.094 | V 0.030<br>Co 0.098 | Cd 0.049 | Ga 0.011<br>Li 0.100 | As 0.024 | V 0.066 | Cr 0.066 | Pb 0.010<br>Sr 0.058 | Cr 0.015<br>Ba 0.060 | As 0.031<br>Li 0.094 | Ni 0.033 |
| 0.1–1 | Ni 0.158 | | | | Cd 0.160 | Pb 0.190 | Ni 0.243 | Co 0.308 | | | | |
| 1–10 | Rb 1.362 | Sr 1.702 | Ba 2.131 | B 5.832 | Rb 1.672 | Sr 5.613 | Ba 6.950 | | Rb 1.029<br>Cu 7.039 | Mn 1.966<br>Si 7.105 | Al 2.578 | B 3.271 |

**Table 1.** *Cont.*

| mg kg$^{-1}$ | BODY (B) | | | | GASTER (G) | | | | BODY without GASTER (B−G) | | |
|---|---|---|---|---|---|---|---|---|---|---|---|
| 10–100 | Cu 10.06<br>Zn 58.40 | Al 12.17<br>Fe 73.06 | Si 24.40 | Mn 49.44 | B 12.31 | Cu 16.49 | Al 28.09 | Si 53.42 | Zn 21.31 | Fe 45.71 | |
| 100–1000 | Na 223.14 | Mg 356.17 | Ca 580.03 | | Zn 132.24<br>Mg 431.15 | Fe 136.97 | Mn 149.25 | Na 273.46 | Na 220.66 | Ca 231.79 | Mg 323.49 |
| >1000 | S 2323.33 | P 3859.63 | K 3964.07 | | Ca 1317.58 | S 1875.13 | K 4294.02 | P 4565.84 | S 2649.64 | P 3675.06 | K 3829.48 |

*(b) Control Area (Perugia)*

In the other six intermediate classes, all the remaining trace elements were distributed.

The results for the most toxic elements highlighted that at Alviano Lake, Hg and As were always more abundant than in the control area, Cd was higher in the control area, and finally Pb was similar between the two areas.

Overall, trace element mean concentrations in Alviano honey bee tissues, compared to the mean values measured in honey bees from the control area, showed significant differences for some elements. These elements are listed below, and between brackets is indicated the ratio between the concentrations measured at Alviano and those at the control area: whole bee body (B): Cd (0.6), Ca (0.9), Mg (1.3), Na (1.4), Sr (1.4), Cs (2.4), Rb (3.1), and As (3.8); gaster (G): Ag (0.4), Cd (0.5); Na (1.6), Cs (2.7), As (3.3), and Rb (4.1); and body without gaster (B−G): Ca (0.9), Na (1.3), Mg (1.3), Cs (2.4), Rb (3.2), and As (4.1).

The ratios of the mean concentrations of the 30 trace elements measured in the gaster to those measured in the body without gaster (G/B−G) were organized into 4 classes (varying by factors of 10), from lower than 1 to higher than 100 (Table 2a,b).

The only elements in the Alviano honey bee tissues with a G/B−G ratio lower than 1 were As (0.60) and S (0.70). Similar values were also found in the control area samples (S, 0.71, and As, 0.75). Ratio values higher than 100 were found only for Sr (107.66) in the Alviano samples, while for the control area, in addition to Sr (104.26), Ba (116.32) also reached this level. Overall, 19 elements showed ratios between 1 and 10 in the Alviano Lake samples. A similar condition was found in the control area, with 20 elements, with the addition of Be and V compared to the Alviano samples, while Ag was instead grouped in the superior class. The class 10–100 ratio in the Alviano samples included 8 elements, and it was differentiated from the control area samples for V and Be (both in the lower class) and Ba (in the superior class), as well as the addition of Ag (Figure 2).

In addition, the G/B−G ratios of the toxic elements for the biota showed lower values in the Alviano Lake samples, in comparison to those from the control area: As (0.60 vs. 0.75), Hg (1.10 vs. 1.91), Pb (13.47 vs. 19.91), and Cd (26.33 vs. 41.76) (Figure 2).

To further evaluate the metal contamination levels in honey bee tissues, we applied the Honeybee Contamination Index (HCI) [40]. In the case study, we observed that it was sufficient to apply only HCI$_2$, the index that uses the lower threshold values, because all the values of this index were negative. In fact, the lower threshold was not exceeded for any element (low contamination condition). The only exception was Ni, for which only 1 sample out of 24 (Alviano site 1, hive b, June 2020) showed a positive HCI$_2$ value (0.020), which indicated an intermediate contamination condition, considering that the relative HCI$_1$ value (higher threshold) was −0.46. The mean HCI (HCI$_{mean}$) of the 4 analysed metals (Alviano Lake: HCI$_{mean1}$ = −1.264, HCI$_{mean2}$ = −0.858; control area: HCI$_{mean1}$ = −1.091, HCI$_{mean2}$ = −0.685) highlighted an overall condition of low contamination. Figure 3 shows that the control areas samples generally manifested greater presences of these metals in bee tissues (i.e., higher HCI$_{mean}$ values).

**Table 2.** Ratios between the mean concentrations of the 30 trace elements measured in the gaster and those measured in the body without the gaster (G/B−G) of the (**a**) Alviano Lake and (**b**) control area (Perugia) samples. The ratios are organized, according to their level, into four classes varying by factors of 10, from lower than 1 to higher than 100.

| (a) Alviano Lake | | | | | | | | | |
|---|---|---|---|---|---|---|---|---|---|
| **Ratio Gaster/Body without Gaster** | **(G/B−G)** | | | | | | | | |
| <1 | As<br>0.60 | S<br>0.70 | | | | | | | |
| 1–10 | Hg<br>1.10<br>Fe<br>2.82 | Li<br>1.13<br>Zn<br>4.63 | P<br>1.25<br>Cr<br>5.55 | K<br>1.31<br>Ca<br>6.09 | Mg<br>1.40<br>B<br>6.53 | Na<br>1.46<br>Si<br>7.29 | Rb<br>2.07<br>Ni<br>7.61 | Ag<br>2.15<br>Ga<br>7.85 | Tl<br>2.55<br>Cs<br>9.91 | Cu<br>2.56 |
| 10–100 | V<br>10.37 | Al<br>10.48 | Be<br>12.22 | Pb<br>13.47 | Cd<br>26.33 | Co<br>28.95 | Mn<br>75.37 | Ba<br>88.22 | | |
| >100 | Sr<br>107.66 | | | | | | | | |
| (b) Control Area (Perugia) | | | | | | | | | |
| **Ratio gaster/body without gaster** | **(G/B−G)** | | | | | | | | |
| <1 | S<br>0.71 | As<br>0.75 | | | | | | | |
| 1–10 | Li<br>1.07<br>B<br>3.80 | K<br>1.12<br>Tl<br>4.70 | Na<br>1.25<br>Ca<br>5.65 | P<br>1.25<br>Zn<br>6.21 | Mg<br>1.34<br>Si<br>7.56 | Rb<br>1.65<br>Ni<br>7.79 | Hg<br>1.91<br>Ga<br>7.89 | Cu<br>2.34<br>Cr<br>8.18 | Fe<br>2.93<br>Cs<br>8.61 | Be<br>3.60<br>V<br>9.39 |
| 10–100 | Al<br>11.18 | Ag<br>14.76 | Pb<br>19.91 | Co<br>35.59 | Cd<br>41.76 | Mn<br>77.73 | | | | |
| >100 | Sr<br>104.26 | Ba<br>116.32 | | | | | | | | |

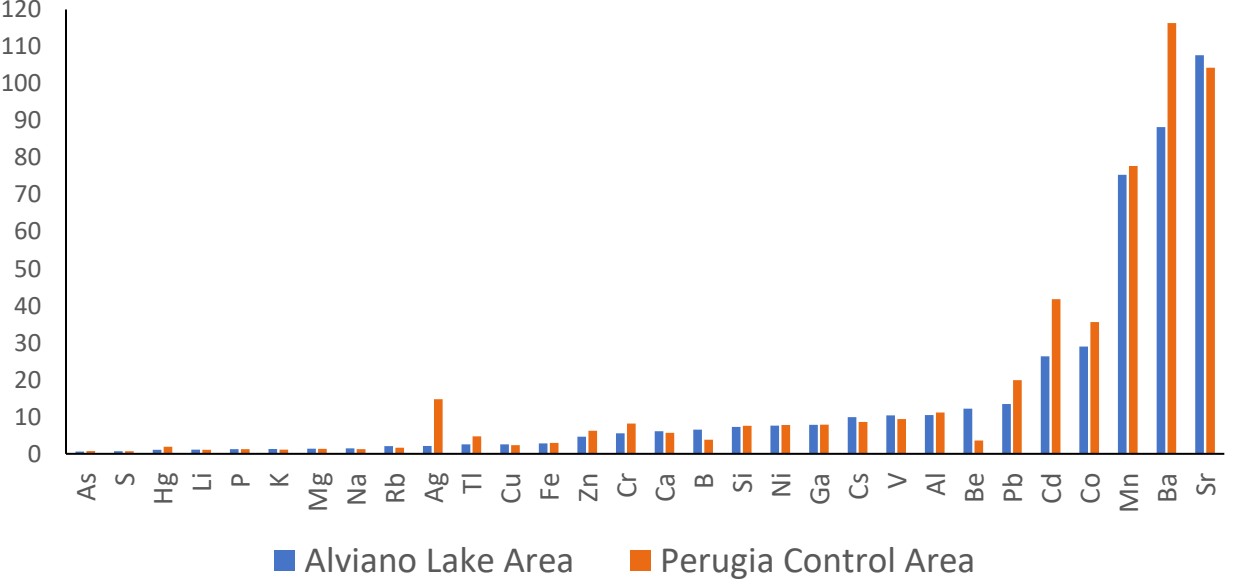

**Figure 2.** Ratios between the mean concentration of the 30 trace elements measured in the gaster and those measured in the body without the gaster (G/B−G) of the Alviano Lake area (blue columns) and Perugia control area (orange columns).

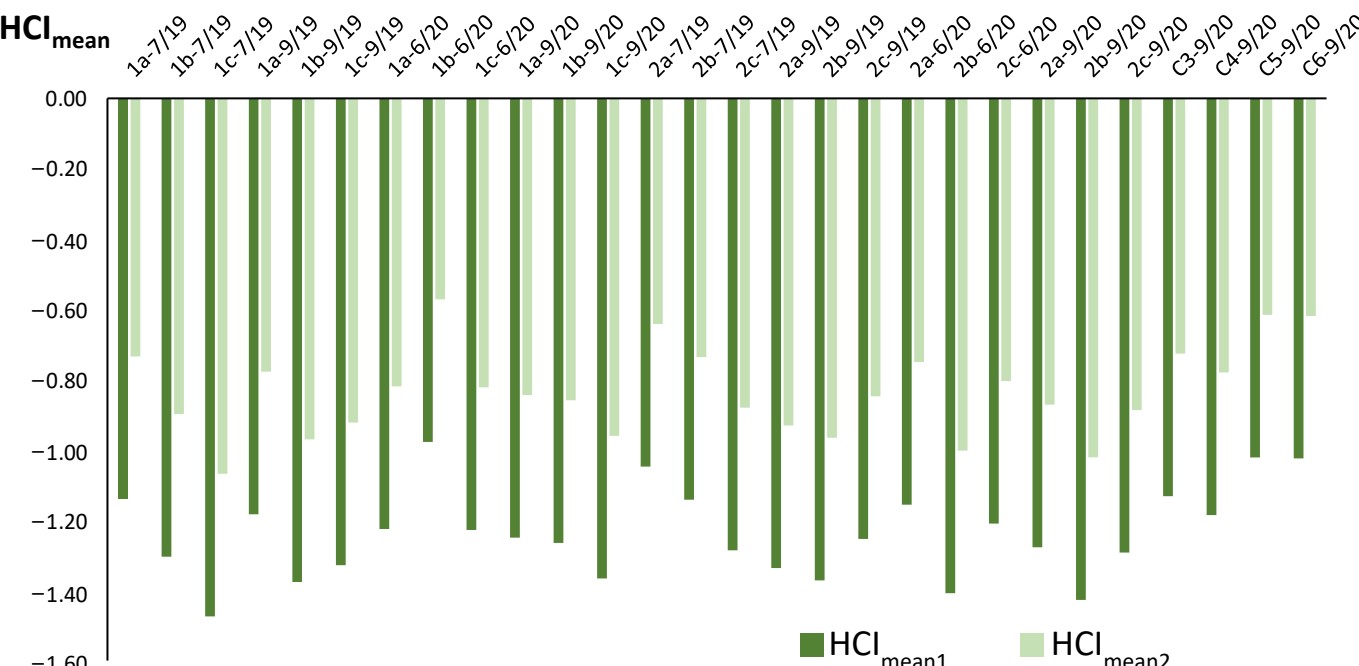

**Figure 3.** Mean HCI (Honeybee Contamination Index) of the four analysed metals (i.e., Cd, Pb, Cr, and Ni); a positive $HCI_1$ indicates a high contamination, a negative $HCI_2$ indicates a low contamination, and finally, when $HCI_1$ is negative and $HCI_2$ is positive, an intermediate contamination level is present. For each sample is indicated the site number: 1, 2 (Alviano Lake sites), C3, C4, C5, C6 (control area (Perugia) sites); the different hives analysed at each site: a, b, c; and the month and year of sample collection.

## 4. Discussion

Our study analysed the concentrations of 30 elements in the tissues of the whole bee body (B), in the gaster, and in the remaining part of the bee body without the gaster. In the body samples, significant differences between the concentrations measured in the Alviano Lake area and in the control area were found for eight elements, of which six elements (Mg, Na, Sr, Cs, Rb, and As) showed higher average values at Alviano Lake, while only two elements (Ca and Cd) had higher concentrations in the control area. The other 22 elements showed a certain degree of similarity between Alviano Lake and the control area, which was chosen as a low-impact zone of metal contamination in the Umbria region of Central Italy [40].

In the literature, studies analysing the concentrations of a high number of elements in the tissues of whole bees are rare. In this regard, the careful investigation of Zarić et al. (2021) [47], conducted at an apiary composed of 21 hives (28 elements measured in 17 specimens per hive, for a total of 357 analyses) in the village of Mesić close to Vršac, Serbia, was taken as a reference. The apiary is located in an agricultural territory connected to a protected area and therefore has environmental characteristics very similar to the agricultural and protected areas of Alviano Lake.

In Mesić, the following decreasing sequence for the elements analysed in the bees' tissues (concentration mg kg$^{-1}$, dw) was found: K (8604.10) > P (7199.10) > S (5434.48) > Ca (1067.19) > Mg (993.52) > Na (933.81) > Fe (139.90) > Zn (104.43) > Mn (83.81) > Al (52.05) > Cu (20.67) > B (6.959 > Sr (4.49) > Ba (4.28) > Rb (3.97) > Cd (0.43) > Mo (0.22) > Cr (0.14) > Co (0.12) > Pb (0.11) > Se (0.08) > V (0.08) > As (0.07) > Li (0.04) > Sb (0.01) > Cs (0.01) > U (0.01) > Tl (0.004). Some elements have been examined only at Alviano Lake, i.e., Si, Ni, Hg, Ga, Ag, and Be, while Mo, Se, Sb, and U are elements only measured in Mesić. A comparison of the study carried out in Mesić with our results (Table 1a) shows a similar sequence of elements, with higher concentration levels in Alviano Lake only for Rb, Tl, As, and Li.

Concerning arsenic in bee tissues, in a subsequent survey by Zarić et al. (2022) [48], conducted as well in the locality of Mesić, again a low average value (0.041 mg kg$^{-1}$) was found, while at other sites subjected to a more significant environmental impact, such as Belgrade and the thermal power plant region (TPP), higher mean values of 0.21 mg kg$^{-1}$ and 0.37 mg kg$^{-1}$ were found, respectively.

The average level of contamination of As found in the bees of Alviano Lake (0.105 mg kg$^{-1}$; Figure 4) is, however, lower than what was found in the study of Van der Steen et al. (2016) [5] conducted throughout the Dutch territory on 150 apiaries (As: average 0.85 ± 0.33, range 0.13–1.64 in mg kg$^{-1}$). In addition, these results [5] were comparable to those found in the previous survey conducted in 2006 at the three Dutch sites of Maastricht (urban area with industries), Buggenum (rural area with a power plant), and Hoek van Holland (port and large industrial area), with an average value of As of 0.71 ± 0.02 mg kg$^{-1}$ [45].

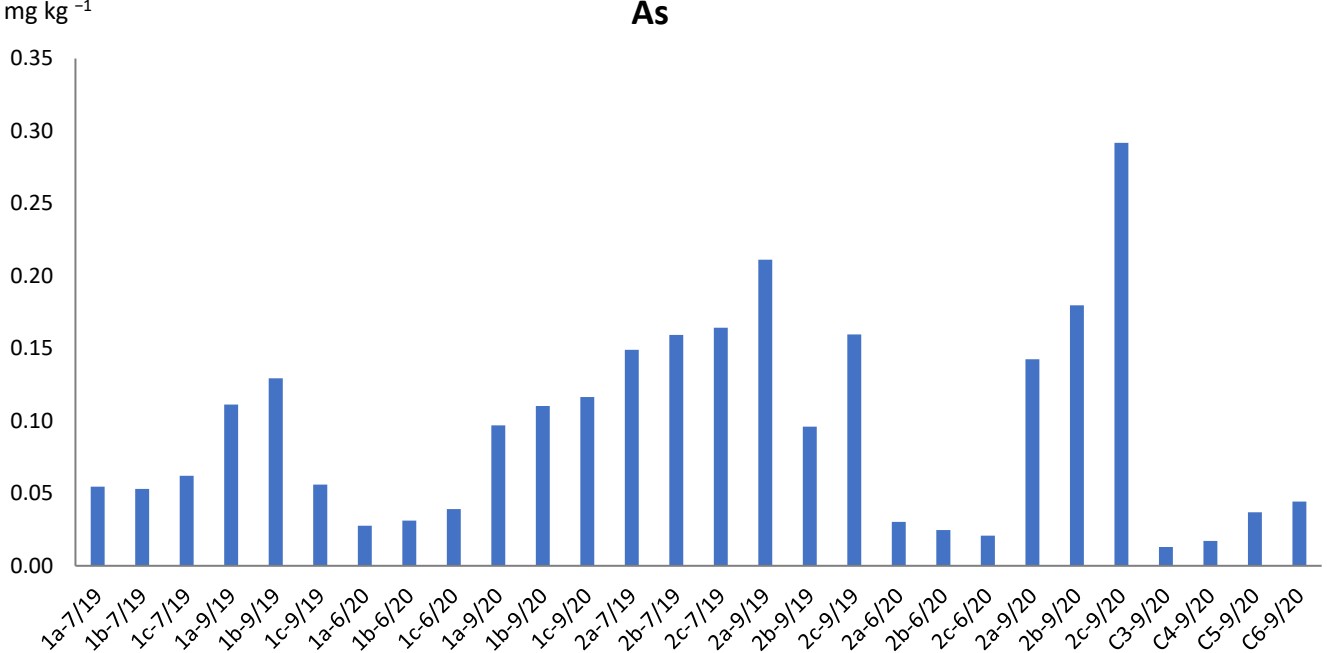

**Figure 4.** Concentration of As in the whole body tissues of honey bees of the Alviano Lake and control area (Perugia) sites (mg kg$^{-1}$ dry weight). For each sample is indicated the site number: 1, 2 (Alviano Lake sites), C3, C4, C5, C6 (control area (Perugia) sites); the different hives analysed at each site: a, b, c; and the month and year of sample collection.

In the study of Grenier et al. (2021) [49] in Québec City (Canada) at six sites (four in urban areas and two in rural areas), arsenic showed a significantly higher value in bee tissues located in urban sites than in rural sites. In fact, from July to September (2019) the average values of As did not exceed 0.08 mg kg$^{-1}$ in rural areas, and they were about 0.15–0.28 mg kg$^{-1}$ in urban areas.

These surveys showed that, compared to the rural area of Alviano Lake, there are usually higher values of As in the tissues of bees in areas with high environmental impact, such as urban areas, while lower values are found in areas with low environmental impact, such as rural sites. The higher levels of arsenic observed in the rural territory of Alviano Lake are due to natural processes of water–volcanic rock interaction present in the territory of Orvieto, of which the area of Alviano Lake is an integral part [50].

The application of the HCI for the four heavy metals (i.e., Cd, Cr, Ni, and Pb) for which in the literature threshold levels are known showed negative mean values at Alviano Lake for HCI$_{mean1}$ (−1.264) and HCI$_{mean2}$ (−0.858), which are significantly lower than the HCI$_{mean}$ values for the sites of the Umbria (Italy) region reported in Goretti et al., 2020 [40] and compared to the metadata of various sites from Central Italy [9,11,22,51,52]

and various European regions [5,12,13,45,53–55]. Therefore, HCI$_{mean}$ showed a peculiar low level of contamination of bee tissues of the rural sites of Alviano Lake for the four metals considered.

With regard to mercury (average value at Alviano Lake area equal to 0.006 mg kg$^{-1}$), in 2014 Zarić et al. (2018) [56], in the locality of Mesić, an agricultural locality of Serbia with characteristics similar to the territory of Alviano Lake, found that the level of mercury in bees amounted to 0.026 mg kg$^{-1}$, a value that corresponds to 4.5 times that found in Alviano bees. In the study by Toth et al., 2016 [57], carried out in eastern Slovakia in the sites of Košice (area of the University of Veterinary Medicine and Pharmacy) and Rozhanovce (small, rural village), the levels of Hg detected in the tissues of bees were respectively about 7 times (0.04 mg kg$^{-1}$) and 1.5 times (0.008 mg kg$^{-1}$) the values found at the sites of Alviano Lake. These results show that mercury contamination in the bees of the Alviano Lake area stands at a limited level, even if it is about three times higher than the values found in the control area. This result is probably influenced by the environmental conditions of its catchment area, subjected until 1980 to the intense mining activity of Hg in the nearby Monte Amiata mining district. However, analysing the individual samples of the Alviano Lake area, a peak of Hg in the sample of July 2019 (Site 1a) was revealed. In this sample, Hg values for the body amounted to 0.049 mg kg$^{-1}$, thus showing a level comparable to the contamination of productive areas (Figure 5).

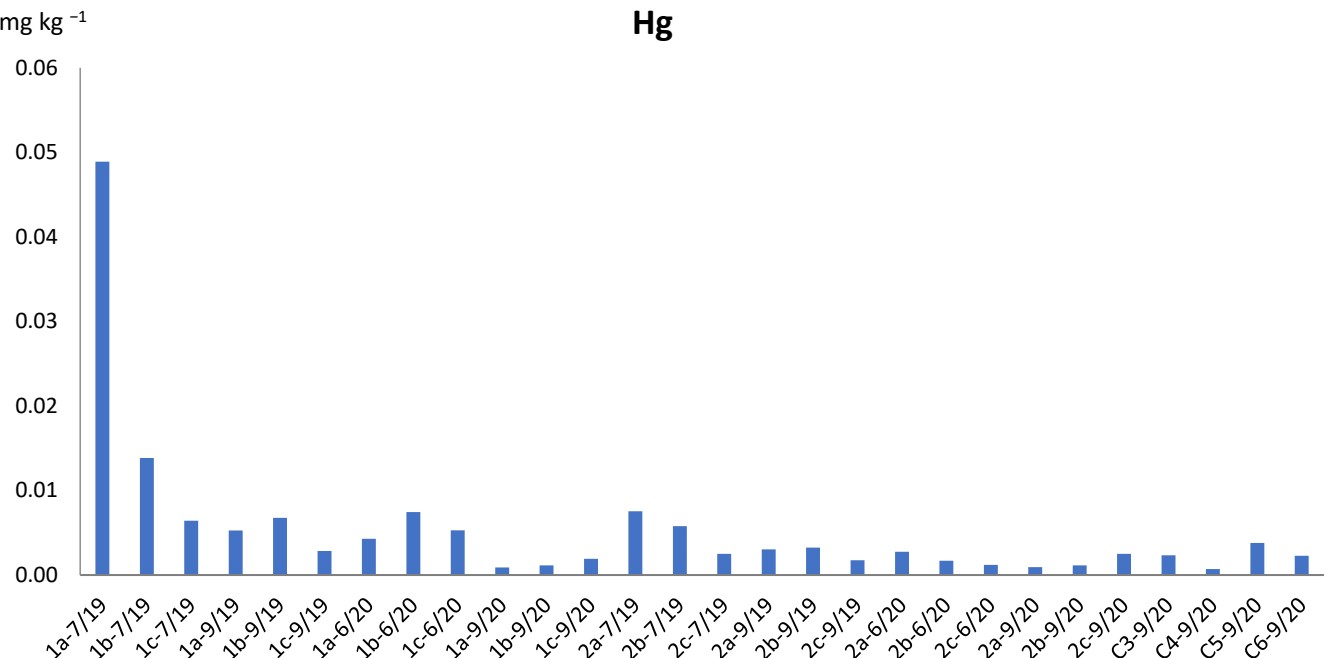

**Figure 5.** Concentrations of Hg in the whole body of honey bees of Alviano Lake and control area (Perugia) sites (mg kg$^{-1}$ dry weight). For each sample is indicated the site number: 1, 2 (Alviano Lake sites), C3, C4, C5, C6 (control area (Perugia) sites); the different hives analysed at each site: a, b, c; and the month and year of sample collection.

Examining the levels of trace elements along the body axis of honey bees of Alviano allowed us to highlight differences in bioaccumulation between the two body parts examined.

From Table 1a it is noted that the comparison of the element sequences among tissues of the body, gaster, and body without gaster showed a greater variation between body and body without the gaster than between body and gaster, confirming that the higher levels of the elements detected in the gaster affected considerably their relative content in the body. In fact, in the tissues of the bees of Alviano Lake, significant differences in the concentrations were found for all the elements analysed (except Li) in the gaster compared to the body without the gaster.

The elements generally showed a ratio of G/B−G > 1 (Table 2a), highlighting that the gaster is the main part of the body for the accumulation of these elements, being the location of most of the bee's digestive system (crop, midgut, and hindgut) and excretory system (Malpighian tubules). For the toxic elements, this is particularly evident for Pb (ratio 13.47) and Cd (ratio 26.33).

Only As (ratio 0.60) and S (ratio 0.70) have a ratio that is <1, while Hg had a ratio of 1.10, indicating a minimum level of variation between these two parts of the bee body.

Arsenic is generally bound to sulphur, with which it forms inorganic and organic arsenic compounds, with different oxidation states [58]. In particular, muscle proteins show the highest sulphur concentrations, especially when they are rich in sulphur amino acids. Sulphur is essential in maintaining the tertiary structures of proteins by giving them stability through the bond of the disulphide bridge. Therefore, given that muscle proteins are mainly present in the muscles of flight of the thorax of insects, the affinity of S and As with the anterior part of the bee body (body without the gaster) is unsurprising. On the other hand, this phenomenon of affinity towards S is also applicable to Hg because it binds to muscle proteins rich in amino acids with the –SH group [59–62]. In Selvaggi et al. (2022) [63], the bioaccumulation of Hg in red swamp crayfish tissues showed significant and selective accumulation in the abdominal muscles compared to the hepatopancreas, an inverse trend compared to all other metals analysed (Cr, Mn, Fe, Co, Ni, Cu, Zn, Ag, Cd, and Pb). These results regarding the details of Hg bioaccumulation in animal tissues were studied mainly in aquatic invertebrates [64], and specifically in the bivalve *Corbicula fluminea* and the crayfish *Astacus astacus* [65], in the crayfish *Pacifastacus leniusculus* [66], in different species from five dragonfly families (Odonata: Anisoptera) [67], and in the copepod *Tigriopus japonicus* [68].

## 5. Conclusions

The study of the bioaccumulation of 30 elements in *Apis mellifera ligustica* from Alviano Lake (Central Italy), a rural and protected natural area, highlighted a low level of contamination in the tissues of these bioindicators.

The concentrations found for the analysed elements could be considered reference models for low-environmental-impact territories. Even mercury, despite the environmental context, showed usually limited contamination levels, except in a single sample. On the other hand, the natural processes of water–volcanic rock interaction influenced the arsenic levels, which showed higher concentrations than those found in bee tissues from other rural areas. However, As concentrations were lower than those measured in samples from urban and productive areas. Other toxic elements, such as Cd and Pb, showed lower levels compared to the low contamination threshold values known in the literature, which were used for the HCI calculation.

The analysis highlighted the extent of the 30 elements analysed in the bee tissues. Be and Ag were the elements with the lowest values, while S, P, and K had the highest values.

The analysis of the elements along the longitudinal axis of the body of the bees showed that the gaster, the location of most of the digestive and excretory systems, showed a greater and more significant presence of these elements compared to the remaining part of the bee body (body without the gaster), in particular for V, Al, Be, Pb, Cd, Co, Mn, Ba, and especially Sr. The only exceptions were As, S, and, to a lesser extent, Hg due to the affinity of As and Hg towards the –SH groups of muscle proteins.

**Supplementary Materials:** The following supporting information can be downloaded at: https://www.mdpi.com/article/10.3390/app13126918/s1. Table S1. (a) Concentrations of the 30 trace elements in the whole honey bee body (B) (mg kg$^{-1}$ dry weight; single values, means, standard deviations, maxima and minima, limit of detection) for Alviano Lake and control area (Perugia) sites. For each sample is indicated the site number: 1,2 (Alviano Lake sites), C3, C4, C5, C6 (control area (Perugia) sites); the different hives analysed at each site: a, b, c; and the month and year of sample collection. (b) Concentrations of the 30 trace elements in the abdominal segments after the petiole (G) (mg kg$^{-1}$ dry weight; single values, means, standard deviations, maxima and minima, limit of detection) for Alviano Lake and control area (Perugia) sites. For each sample is indicated the site number: 1,2 (Alviano Lake sites), C3, C4, C5, C6 (control area (Perugia) sites); the different hives analysed at each site: a, b, c; and the month and year of sample collection. (c) Concentrations of the 30 trace elements in the remaining part of the bee body (B−G) (mg kg$^{-1}$ dry weight; single values, means, standard deviations, maxima and minima, limit of detection) for the Alviano Lake and control area (Perugia) sites. For each sample is indicated the site sampled: 1, 2 (Alviano Lake sites), C3, C4, C5, C6 (control area (Perugia) sites); the different hives analysed at each site: a, b, c; and the month and year of sample collection.

**Author Contributions:** Conceptualisation, E.G., M.P., G.L.P., A.C.E., T.G., C.P., P.G., F.B., R.S. and D.C.; data curation, E.G., M.P., G.L.P., C.P., F.B., R.S. and D.C.; methodology, E.G., M.P., G.L.P., A.C.E., T.G., C.P., P.G., F.B., R.S. and D.C.; investigation, E.G., M.P., G.L.P. and P.G.; writing—original draft, E.G., M.P., R.S. and D.C.; writing—review and editing, E.G., M.P., R.S. and D.C. All authors have read and agreed to the published version of the manuscript.

**Funding:** This research was funded by the Ministero Istruzione dell'Università e della Ricerca (MIUR) and the University of Perugia through the program "Dipartimenti di Eccellenza 2018–2022" (grant AMIS).

**Institutional Review Board Statement:** Not applicable.

**Informed Consent Statement:** Not applicable.

**Data Availability Statement:** The data presented in this study are available in the Supplementary Materials.

**Acknowledgments:** We thank the Alviano Lake WWF Oasis staff members Alessio Capoccia, Marianeve Medori, and Laura Saenz De Buruaga for their great availability; special thanks go to Alessio for his fundamental support in the sampling over the years of research. We thank the beekeepers Francesco Bartocci, Andrea Sportoletti, and Stefania Palmerini for supporting the sampling campaign by making their apiaries available as the control area of the Perugia territory.

**Conflicts of Interest:** The authors declare no conflict of interest.

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
