# Peer review of "Bioaccumulation of Trace Elements along the Body Longitudinal Axis in Honey Bees"

_applsci, doi:10.3390/app13126918_

Round 1

Reviewer 1 Report

The study was conducted with aim to identify the levels of trace element accumulation in the honey bees from artificial basin “Lake Alviano” as the environment rich in Hg and correlate with trace levels in samples from control location. The study is very interesting and with great scientific impact in the field. The manuscript is well written ad adequate methodology was applied. However, certain details need to be added mostly in the Material and Method section.

Line 46: “One of the main responsible factors is the use of neonicotinoid pesticides”

Please moderate this statement, considering that it is the statement of several researchers. e. g. According to some authors one of the main responsible factors is the use of neonicotinoid pesticides.

Line 72: Please explain “WWF” abbreviation.

Lines 108-110: Add more details about sampling of bees. How you sampled them, e.g. at the entrance of the hive or collecting them from the frames, from which frames, peripheral or central etc.

What does this mean: “Honey bee sampling was carried out according to the safety rules and respecting the colonies, avoiding disturbing the insect activity”. Which safety rules, add reference for this? How exactly you “avoided disturbing of insect activities”?

What was the beekeeping practice in both sites, were the colonies treated with anything? Add all the details.

Lines 187-189: How you reached 24 samples, if collected from 3 hives on two sites (6 samples) in two years (6 x 2 = 12)

Lines 190-193: “of each apiary at both Alviano sampling sites and between the two apiaries, both at site 1 and site 2.”

This paragraph is confusingly written. You are talking about different apiaries and different sites. Were there more apiaries on the sites? Please uniform this with the paragraph in the Material and Methods section where you mention only three hives from each of the two sites (not mentioning different apiaries on each site).

Consider adding of graphs. It is hard to focus on the tables and textual description.

Author Response

Comments and Suggestions for Authors

The study was conducted with aim to identify the levels of trace element accumulation in the honey bees from artificial basin “Lake Alviano” as the environment rich in Hg and correlate with trace levels in samples from control location. The study is very interesting and with great scientific impact in the field. The manuscript is well written ad adequate methodology was applied. However, certain details need to be added mostly in the Material and Method section.

Authors’ response: We thank the reviewer for their precious and constructive comments.

Line 46: “One of the main responsible factors is the use of neonicotinoid pesticides”

Please moderate this statement, considering that it is the statement of several researchers. e. g. According to some authors one of the main responsible factors is the use of neonicotinoid pesticides.

Authors’ response: We modified the statement as suggested, now the sentence is “According to some authors, one of the main responsible factors is the use of neonicotinoid pesticides”.

Line 72: Please explain “WWF” abbreviation.

Authors’ response: We explained the acronym WWF, now the sentence is “…a naturalistic oasis of the World Wide Fund for Nature (WWF).”

Lines 108-110: Add more details about sampling of bees. How you sampled them, e.g. at the entrance of the hive or collecting them from the frames, from which frames, peripheral or central etc.

Authors’ response: We integrated details about the sampling of bees. Now the sentences is “About 100 forager bees were collected at the entrance of each hive with a plastic bag”.  

What does this mean: “Honey bee sampling was carried out according to the safety rules and respecting the colonies, avoiding disturbing the insect activity”. Which safety rules, add reference for this? How exactly you “avoided disturbing of insect activities”?

Authors’ response: We added some explanation about these statements. Now the sentence is “Honey bee sampling was carried out according to the good beekeeping practices, avoiding to interfere with the colony activity through opening the hives”.

What was the beekeeping practice in both sites, were the colonies treated with anything? Add all the details.

Authors’ response: We integrated details about the beekeeping practices in both sampling areas. Now the sentences is “All the apiaries sampled were characterized by treatment protocols compatible with organic beekeeping practices.”.  

Lines 187-189: How you reached 24 samples, if collected from 3 hives on two sites (6 samples) in two years (6 x 2 = 12)

Authors’ response: In the Alviano Lake area we sampled 3 different hives for each of the two sites, two times a year, for two years (2019 and 2020). We explained both in Materials and Methods and in the Results sections the exact number of samples analyzed for both sampling areas. Specifically, we integrated the required information in Materials and Methods in the following sentences: “The survey was carried out in 2019 and 2020 at Alviano Lake (sites 1 and 2), with two annual honey bee samplings during the warm season (July and September 2019, June and September 2020) (Figure 1). Three hives (a, b, and c) were sampled at each of the two Alviano sites (site 1: 70 hives; Site 2: 64 hives), taking care to select hives distant from each other (one in the centre and the other two at the ends of the apiary).”, and “Therefore, for each sample is indicated the site number: 1, 2 (Alviano Lake sites), C3, C4, C5, C6 (Control Area – (Perugia) sites); the different hive analysed at each site: a, b, c; month and year of sampling collection”. In the Results “The survey on Alviano Lake (sites 1, 2), carried out in 2019 and 2020, was based on 72 honey bee samples (24 for each sample type, i.e., B, B-G, and G) collected in four sampling occasions at three different hives for each site”; and “At the four control sites (sites C3-C6), located close to the city of Perugia (Umbria, Central Italy), 12 samples (4 for each sample type, i.e., B, B-G, and G), collected in September 2020, were processed comprehensively as Perugia Control area.”

Lines 190-193: “of each apiary at both Alviano sampling sites and between the two apiaries, both at site 1 and site 2.”

This paragraph is confusingly written. You are talking about different apiaries and different sites. Were there more apiaries on the sites? Please uniform this with the paragraph in the Material and Methods section where you mention only three hives from each of the two sites (not mentioning different apiaries on each site).

Authors’ response: We reformulated the paragraph as suggested. Now it is “The element concentrations in the whole bee tissues (body) of Alviano Lake were not significantly different (Mann-Whitney’s U test, p>0.05) among the three hives (a, b, c), both at site 1 and site 2. No significant differences were detected also between the two sites (1, 2). As a result, all the Alviano samples were subsequently processed comprehensively as Alviano Lake area.”

Consider adding of graphs. It is hard to focus on the tables and textual description.

Authors’ response: We added a new figure (Figure 2), that displays clearly the data presented in Table 2. This figure shows the ratios between the mean concentration of the thirty trace elements measured in the gaster and those measured in the body without the gaster (G / B−G) of the Alviano Lake area and Perugia Control Area.

Reviewer 2 Report

Comments on the manuscript titled: “Bioacumulation of Trace Elements Along the Body Longitudinal Axis in Honey Bees” by Goretti et al. submitted to Applied Sciences mdpi.

The submitted work for evaluation is an interesting ecotoxicological work based on environmental samples. The work concerns a highly sensitive bioindicator (Apis mellifera) on which the functioning of many ecosystems depends.  The results could be extremely important for bee conservation.  It fully deserves to be published in Applied Sciences.

However, the paper needs a few minor improvements.

General comments:

1.               The Authors write: “We hypothesise that this selectivity maybe due to the affinity of these elements with S, which is abundant in the proteins of the flight muscles in the insect thorax, rich in amino acids containing the -SH group. (lines: 20-22)” Unfortunately, the relationships of toxic metals with sulphur are discussed mainly on vertebrate examples (lines: 398-400). From invertebrates an example is given: red swamp crayfish. These need to be discussed more extensively with examples of other insects and invertebrates.

2.               The studies were carried out using an Inductively Coupled Plasma Mass Spectrometer, which requires a table of recoveries to be given in the text. Please add Validation of the analytical method: detection limit and recoveries for the studied elements to manuscript text.  Is that the mentioned RSDs (line153) which is as high as 13%?

3.               The authors focus too much on biogenic elements such as : P, K, Ca, S which have high concentration levels in all studies on living organisms. Instead of that, more space should be devoted to comparing their data to literature information on toxic elements. The discussion is very poor in this respect.

4.               Following other ecotoxicological work, the authors should calculate the correlations between elements and discuss this problem on the basis of the literature.  It would also be useful to carry out a Hierarchical Agglomerative Cluster Analysis (HACA) and on its basis analyse possible sources of contamination as is done in many ecotoxicological works. This would have been extremely useful for the conclusions of this paper, especially as the authors do not consider at all in the paper the potential sources of contamination of the bee specimens they examined.

Technical issues:

1.     Please do not divide words in the title

2.     Line 16-17: Hg showed low contamination levels, with exception of a sample. ….something is not finished here

3.     Please delete a full stop (.) at the end of Keywords

4.     Line 52: remove  “of these elements”

5.     Line 72: WWF please add explanation to abbreviation

6.     Figure 1. Add escription of the pictures glued from the left side. Add names Perrugia and And Alviano lake to the left side bottom graphics.

7.     Materials and Methods: Please add names of producers and country of origin when writing about used  materials/ chemicals.
It is not clear how many samples were involved to studies (both Materials and Methods as well as Results lines 186-193). Please state straightforward

8.     Line 173 “which” instead of “whose”

9.     Line 200 rewrite “ was made organising them ….”

10.  Line 270 No need to repeat about HCI Index, it was already mentioned in Methods

11.  Please format all Tables to mdpi style

12.  Marking of samples from figures 2, 3 and 4 are not explained in Methods section

13.  Check the formatting of references, pay attention to 29, 40, 57,60,..etc…

Quality of language -fine, minor corrections

Author Response

Comments and Suggestions for Authors

Comments on the manuscript titled: “Bioaccumulation of Trace Elements Along the Body Longitudinal Axis in Honey Bees” by Goretti et al. submitted to Applied Sciences mdpi. The submitted work for evaluation is an interesting ecotoxicological work based on environmental samples. The work concerns a highly sensitive bioindicator (Apis mellifera) on which the functioning of many ecosystems depends.  The results could be extremely important for bee conservation.  It fully deserves to be published in Applied Sciences.

However, the paper needs a few minor improvements.

Authors’ response: We thank the reviewer for their precious and constructive comments.

General comments:

  1. The Authors write: “We hypothesise that this selectivity maybe due to the affinity of these elements with S, which is abundant in the proteins of the flight muscles in the insect thorax, rich in amino acids containing the -SH group. (lines: 20-22)” Unfortunately, the relationships of toxic metals with sulphur are discussed mainly on vertebrate examples (lines: 398-400). From invertebrates an example is given: red swamp crayfish. These need to be discussed more extensively with examples of other insects and invertebrates.

Authors’ response: We added some discussion regarding the relationships of toxic metals with sulphur, giving examples of studies involving other invertebrates and insects. Specifically, we added the following sentence: “These results regarding the way of Hg bioaccumulation in animal tissues were studied mainly in aquatic invertebrates [64], and specifically in the bivalve Corbicula fluminea and the crayfish Astacus astacus [65], in the crayfish Pacifastacus leniusculus [66], in different species from five five dragonfly families (Odonata: Anisoptera) [67], and in the copepode Tigriopus japonicus [68].”

  1. The studies were carried out using an Inductively Coupled Plasma Mass Spectrometer, which requires a table of recoveries to be given in the text. Please add Validation of the analytical method: detection limit and recoveries for the studied elements to manuscript text.  Is that the mentioned RSDs (line153) which is as high as 13%?

Authors’ response: We added the required information on the validation of the analytical method. Specifically, for the recoveries, we added this sentence: “The accuracy of the method was obtained using standard reference materials (BCR-185R bovine liver). The metal concentrations agreed with the certified value, and recovery fell in the range of 80–120% [40]”; we also added the detection limits “Detection limits of the method (in mg kg−1) were: Ag, 0.0002; Al, 0.08; As, 0.001; B, 0.10; Ba, 0.001; Be, 0.0001; Ca, 0.28; Cd, 0.0002; Co, 0.0005; Cr, 0.003; Cs, 0.0008; Cu, 0.001; Fe, 0.01; Ga, 0.0006; Hg, 0.0007; K, 0.38; Li, 0.09; Mg, 0.03; Mn, 0.003; Na, 0.18; Ni, 0.0004; P, 0.04; Pb, 0.002; Rb, 0.001; S, 0.04; Si, 0.05; Sr, 0.001; Tl, 0.0001; V, 0.0005; Zn, 0.09”. The RSDs mentioned (2.6–13.0 %) is the relative standard deviation calculated by performing three replicate analyses of two multi-element standards solutions, to assess the experimental repeatability.

  1. The authors focus too much on biogenic elements such as: P, K, Ca, S which have high concentration levels in all studies on living organisms. Instead of that, more space should be devoted to comparing their data to literature information on toxic elements. The discussion is very poor in this respect.

Authors’ response:

We thank the reviewer for their comment. The most toxic elements (As, Hg, Cd, and Pb) are treated both in the results and in the discussions. In particular, in the discussions a particular focus is made for As and Hg, through comparisons with other studies in literature. In addition, for the other two toxic elements (i.e., Pb and Cd), the HCI index has been applied and discussed, this index is based on threshold values derived from numerous studies in the literature.

However, the overall levels of the most toxic elements found in this study were always very low, therefore, considering also the fact that the primary outputs of our research were to understand how the trace elements were distributed in the various body parts of the honey bee, and to provide a relevant reference database to characterize low environmental impact territories, we did not consider expanding further the discussion of these toxic elements. However, to give a more complete picture of the contamination by toxic elements, in the conclusions, besides As and Hg, we added the following sentence about Cd and Pb: "Other toxic elements, as Cd and Pb, showed lower levels respect to the low contamination Threshold values known in the literature, used for the HCI calculation".

  1. Following other ecotoxicological work, the authors should calculate the correlations between elements and discuss this problem on the basis of the literature.  It would also be useful to carry out a Hierarchical Agglomerative Cluster Analysis (HACA) and on its basis analyse possible sources of contamination as is done in many ecotoxicological works. This would have been extremely useful for the conclusions of this paper, especially as the authors do not consider at all in the paper the potential sources of contamination of the bee specimens they examined.

Authors’ response:

We thank the reviewer for their suggestion, but we could not include the calculations of correlation or hierarchization between the elements, because this was out of the main output of our study, that did not consider the contamination sources, except for As and Hg. This exception is justified by the values of these two contaminants, that were higher at the Alviano Lake than the control area. In fact, the focus of our research was to understand how the trace elements were distributed in the various body parts (body, gaster, and body without the gaster) of the selected bioindicator (honey bee), and to provide a relevant reference database to characterize low environmental impact territories to be used as a comparison for sites at different levels of impact. This database may help to understand how the relationship between the two parts of the bee body can vary in relation to the pollution level, and in the future reference thresholds for this ratio may be also identified, to better characterise environmental contamination based on honey bee monitoring.

Technical issues:

  1. Please do not divide words in the title

Authors’ response: We modified the formatting of the title as suggested.

  1. Line 16-17: Hg showed low contamination levels, with exception of a sample. ….something is not finished here

Authors’ response: We modified the sentence, now it is “On the other hand, despite the environmental context, Hg showed limited contamination levels, with the exception of a single sample”.

  1. Please delete a full stop (.) at the end of Keywords

Authors’ response: We deleted the full stop as suggested.

  1. Line 52: remove  “of these elements”

Authors’ response: We removed the words “of these elements” as suggested.

  1. Line 72: WWF please add explanation to abbreviation

Authors’ response: We explained the acronym WWF, now the sentence is “…a naturalistic oasis of the World Wide Fund for Nature (WWF).”

  1. Figure 1. Add escription of the pictures glued from the left side. Add names Perrugia and And Alviano lake to the left side bottom graphics.

Authors’ response: We added a description of all the parts of Figure 1. We also modified the figure adding the names of the two areas (Alviano Lake and Perugia Control Area) in the left side bottom graphic.

  1. Materials and Methods: Please add names of producers and country of origin when writing about used materials/chemicals. It is not clear how many samples were involved to studies (both Materials and Methods as well as Results lines 186-193). Please state straightforward

Authors’ response: We added the names of the producers and the country of origin for all the materials and chemicals used.

In the Alviano Lake area we sampled 3 different hives for each of the two sites, two times a year, for two years (2019 and 2020). We explained both in Materials and Methods and in the Results sections the exact number of samples analyzed for both sampling areas. Specifically, we integrated the required information in Materials and Methods in the following sentences: “The survey was carried out in 2019 and 2020 at Alviano Lake (sites 1 and 2), with two annual honey bee samplings during the warm season (July and September 2019, June and September 2020) (Figure 1). Three hives (a, b, and c) were sampled at each of the two Alviano sites (site 1: 70 hives; Site 2: 64 hives), taking care to select hives distant from each other (one in the centre and the other two at the ends of the apiary).”; and “Therefore, for each sample is indicated the site number: 1, 2 (Alviano Lake sites), C3, C4, C5, C6 (Control Area – (Perugia) sites); the different hive analysed at each site: a, b, c; month and year of sampling collection”. In the Results “The survey on Alviano Lake (sites 1, 2), carried out in 2019 and 2020, was based on 72 honey bee samples (24 for each sample type, i.e., B, B-G, and G) collected in four sampling occasions at three different hives for each site”; and “At the four control sites (sites C3-C6), located close to the city of Perugia (Umbria, Central Italy), 12 samples (4 for each sample type, i.e., B, B-G, and G), collected in September 2020, were processed comprehensively as Perugia Control area.”

  1. Line 173 “which” instead of “whose”

Authors’ response: We replaced the word “whose” as suggested.

  1. Line 200 rewrite “ was made organising them ….”

Authors’ response: We rewrote the sentence as required, now it is “The mean concentrations of the thirty trace elements […] for Alviano and control areas were grouped in eight classes…”

  1. Line 270 No need to repeat about HCI Index, it was already mentioned in Methods

Authors’ response: We deleted the repetition regarding the HCI index.

  1. Please format all Tables to mdpi style

Authors’ response: We formatted all the Tables according to MDPI style.

  1. Marking of samples from figures 2, 3 and 4 are not explained in Methods section

Authors’ response: We added in the Materials and Methods section the missing information for the sample codes reported in Figures 3, 4 and 5. Specifically, we added two sentences: “two annual honey bee samplings during the warm season (July and September 2019, June and September 2020)”, and “Therefore, for each sample is indicated the site number: 1, 2 (Alviano Lake sites), C3, C4, C5, C6 (Control Area – (Perugia) sites); the different hive analysed at each site: a, b, c; month and year of sampling collection”.

  1. Check the formatting of references, pay attention to 29, 40, 57, 60,..etc…

Authors’ response: We checked the formatting of all the references as suggested.

Comments on the Quality of English Language

Quality of language -fine, minor corrections

Round 2

Reviewer 1 Report

The authors addressed all the issues and manuscript can be published in present form.